# Shortening Stabilization Time Using Pressurized Air Flow in Manufacturing Mesophase Pitch-Based Carbon Fiber

**DOI:** 10.3390/polym11121911

**Published:** 2019-11-20

**Authors:** Hiroki Shimanoe, Seunghyun Ko, Young-Pyo Jeon, Koji Nakabayashi, Jin Miyawaki, Seong-Ho Yoon

**Affiliations:** 1Interdisciplinary Graduate School of Engineering Sciences, Kyushu University, Fukuoka 816-8580, Japan; 3ES17003K@s.kyushu-u.ac.jp (H.S.); nakabayashi@cm.kyushu-u.ac.jp (K.N.); miyawaki@cm.kyushu-u.ac.jp (J.M.); yoon@cm.kyushu-u.ac.jp (S.-H.Y.); 2Carbon Industry Frontier Research Center, Korea Research Institute of Chemical Technology (KRICT), Daejeon 34114, Korea; feat@krict.re.kr; 3Advanced Materials and Chemical Engineering, University of Science and Technology (UST), Daejeon 34113, Korea; 4Institute for Materials Chemistry and Engineering, Kyushu University, Fukuoka 816-8580, Japan

**Keywords:** carbon fiber, mechanical properties, mesophase pitch, oxidation–stabilization, pressurized air flow

## Abstract

Oxidation–stabilization using pressurized air flows of 0.5 and 1.0 MPa could successfully shorten the total stabilization time to less than 60 min for manufacturing mesophase pitch-based carbon fibers without deteriorating mechanical performance. Notably, the carbonized fiber heat-treated at 1000 °C for 30 min, which was oxidative–stabilized at 260 °C without soaking time with a heating rate of 2.0 °C/min using 100 mL/min of pressurized air flow of 0.5 MPa (total stabilization time: 55 min), showed excellent tensile strength and Young′s modulus of 3.4 and 177 GPa, respectively, which were higher than those of carbonized fiber oxidation–stabilized at 270 °C without soaking time with a heating rate of 0.5 °C/min using 100 mL/min of atmospheric air flow (total stabilization time: 300 min). Activation energies for oxidation reactions in stabilization using pressurized air flows were much lower than those of oxidation reactions using atmospheric air flow because of the higher oxidation diffusion from the outer surface into the center part of pitch fibers for the use of the pressurized air flows of 0.5 and 1.0 MPa than the atmospheric one. The higher oxygen diffusivities resulted in a more homogeneous distribution of oxygen weight uptake across the transverse section of mesophase pitch fibers, and allowed the improvement of the mechanical properties.

## 1. Introduction

Carbon fibers (CFs) have attracted substantial attention again owing to their novel applications in the automotive, windmill, and other energy-saving industries [1,2]. The potential to reduce the weight of car-body frames is one of the issues attracting the most interest to achieve the widespread adoption of electric vehicles (EVs) and plug-in hybrid electric vehicles (PHEVs) [1]. To use CF composites for the reduction of car-body weight, two essential criteria need to be met: a reasonable price and appropriate mechanical performance. For the effective application of CFs to car-body frames, Jim deVries at Ford Motor Company recommended CF mechanical performance regarding tensile strength, elongation ratio, and Young′s modulus of at least 1.7 GPa, 1.5%, and 170 GPa, respectively, and a price of at most $10/kg [1]. In terms of these criteria for the required mechanical performance, some commercial CFs have already reached these thresholds. Commercialized polyacrylonitrile-based carbon fiber (PAN-CF) and mesophase pitch-based carbon fiber (MPCF) fully meet the requirements to be substitutes for steel and aluminium in many applications [3,4]. However, isotropic pitch-based carbon fiber (IPCF) still does not achieve sufficient mechanical performance, exhibiting much lower mechanical performance with a tensile strength of 0.5–1.0 GPa and a Young’s modulus of 30–50 GPa [5,6]. Over the last two decades, many studies have been carried out with the aim of achieving sufficient mechanical performance of IPCF using inexpensive resources such as biomass (cellulose and lignin) and coal and petroleum by-products [7,8,9,10,11,12,13]. Although a few studies obtained suitable mechanical performance of IPCF using inexpensive raw materials [14], it has still not reached a stage suitable for commercialization. For example, Kim et al. reported the successful development of a novel IPCF that has a satisfactory level of tensile properties applicable to CF-reinforced plastics (CFRPs) for car bodies using mixed raw materials of cycled fuel oil and coal tar pitch through a new bromination–dehydrobromination reaction [14,15,16]. In this context, the high price is a major obstacle to adopting CF for CFRP applications for car-body frames.

CFs are produced through a manufacturing process that includes several important stages, such as selection and purification of raw material, preparation of CF precursor (PAN or pitch), spinning, oxidative stabilization, carbonization and further heat treatment of graphitization, and surface treatment [17,18,19]. To reduce the cost of CF production, both improvements in the manufacturing process and reductions in the cost of CF precursor should be pursued. The selection of inexpensive raw material and simplification of the purification and preparation processes of the CF precursor are important factors to reduce the final cost. The selection of inexpensive raw material is particularly important. In this regard, considerable efforts have been made in the last two decades to prepare spinnable precursors from inexpensive resources, the price of which is usually less than $1.0/kg, such as polyolefin [20,21], biomass including lignin [7,8,9,10], ethylene bottom oil [14,15,16], and hyper coal (HPC) [13,22]. The removal of impurities in raw materials that are mainly composed of insoluble inorganic oxides and coke particles was the original reason for nucleating CF surface defects in the carbonization process through catalytic gasification [23]. However, no notable progress has been made in simplifying the purfication process during the last two decades. Nonetheless, this could be a promising way to select highly purified raw materials such as HPC for reducing the production cost of the precursor. Recently, Yang et al. reported interesting novel preparation methods for obtaining spinnable pitch precursors using HPC as an inexpensive raw material [13,22,23]. Preparation and carbon yields of precursors are also important issues. Mesophase and isotropic pitch precursors are more attractive than PAN because they usually exhibit a much higher fixed carbon yield. Additionally, for pitch precursors, there is the advantage that no hazardous gas emissions occur in carbonization.

In the manufacturing process, development of the carbonization and graphitization processes has almost been completed. However, there is still much room for improvement in the oxidation–stabilization process, because this process remains the least efficient, most time-consuming, and costliest [18,24]. In the manufacture of pitch-based CFs, melt spun pitch fibers are usually converted from a thermoplastic solid state to thermosetting through the oxidation-crosslinking reactions of pitch-constituting molecules in the stabilization stage. Conventional oxidation–stabilization employs thermal oxidation using a sufficient amount of atmospheric air flow at a temperature range of 150–300 °C and a long duration of a few hours [24]. Thus, it is one of the most important tasks to shorten the total stabilization time to reduce the cost of manufacturing CFs. However, shortening of the stabilization time, that is, performing rapid oxidation at high temperatures, usually reduces the mechanical performance, especially tensile strength, through the formation of a heterogeneously oxidized state in the transverse section of pitch fibers. Thus, stabilization should proceed slowly to confer optimal and homogeneous distribution of the amounts of oxygen weight-gain (uptake) on stabilized fibers across their transverse section; thus, a long stabilization time at a relatively low temperature is required.

Many studies have been conducted to optimize the mechanical properties of CFs in terms of air oxidation conditions in the stabilization process [24,25,26]. For example, Mochida et al. measured the oxygen distribution in the transverse section of a mesophase pitch fiber after stabilization using electron probe X-ray microscopy and secondary ion mass spectrometry. On the basis of these measurements, they reported that homogeneous oxidation across the transverse section of pitch fibers is one of the most important factors to obtain high tensile strength in the resultant MPCF [25]. Moreover, Yoon et al. reported that the slower the oxidation–stabilization of mesophase pitch fibers, the better the tensile strength achieved, because slower oxidation resulted in less oxidation decomposition of pitch molecules and more homogeneous oxidation across the fiber cross section [26].

Despite the accumulation of such studies during the last three decades, the appropriate tensile strength of the carbonized MPCF of over 2.5 GPa through oxidation–stabilization with a total oxidation–stabilization time of less than 1 h and successive carbonization at a relatively low temperature of 1000 °C has not been achieved, because the slow diffusion rate of oxygen molecules into the center part of pitch fiber and its low oxidation reactions always prevent optimized homogeneous stabilization across the transverse section of pitch fibers within a short time. The main reason for such a slow diffusion rate of oxygen may be the compact alignment of structural units, that is, stacking clusters and domains, of pitch materials [27,28]. Yang et al. estimated that the average radii of free volumes on various mesophase pitches and their fibers were in the ranges of 0.24–0.25 nm and 0.25–0.26 nm, respectively [29]. The average kinetic radii of oxygen and nitrogen are 0.17 and 0.19 nm, respectively, which means that it is very difficult for the effective air diffusion to occur for optimized oxidation reactions in mesophase pitch fibers in conventional atmospheric stabilization. In this context, the fast oxygen diffusion from the outer surface to the center part of the pitch fiber and its rapid and homogeneous oxidation of pitch fibers should be key to rapidly optimizing stabilization for high mechanical performance of the resulting mesophase pitch-based CFs. 

Some pioneering studies of the oxidation–stabilization of various types of mesophase pitch including fibrous form under a mild pressure of oxygen or air flow have been carried out [30,31,32,33]. Cornec et al. and Fathollahi et al. reported in their papers that the oxidation–stabilization of mesophase pitch fibers under a moderate oxygen pressure can be effective in raising the amounts of oxygen uptake and increasing the stabilization depths significantly, even at a low temperature of 170 °C [30,31]. Further, such lower oxidative temperatures under mild pressure of oxygen could afford more effective penetration of stabilization depth, lessening destructive effects of over-oxidation, and potentially shorter processing times [31]. Liedtke et al. and Tseng et al. also studied the oxidation of mesophase pitch particles under the flow of pressurized air flow, and they reported the same effects on the oxidation of mesophase pitch [32,33].

In this study, we aimed to reduce the total stabilization time to less than 1 h without causing deterioration of the mechanical performance of the resultant MPCFs. We aimed for the target mechanical performance of tensile strength and Young′s modulus of the carbonized fiber to be equal to the mechanical performance of the carbonized fiber, which could be prepared using the same precursor pitch fiber, oxidative–stabilized at 270 °C with the heating temperature of 0.5 °C/min using atmospheric 100 mL/min of air flow, and carbonized at 1000 °C for 30 min. To achieve this goal, we attempted oxidation–stabilization of mesophase pitch fibers at the temperature range of 250–270 °C using the pressurized air flows of 0.5–1.0 MPa. For comparison, we also performed oxidation–stabilization of the same mesophase pitch fibers using atmospheric air flow. We carried out thermo-gravimetric analysis (TGA) to analyze the oxidation reactions in the stabilization kinetically.

## 2. Materials and Methods 

### 2.1. Material

AR mesophase pitch (ARMP) was supplied by Mitsubishi Gas Chemical Company, Tsukuba, Japan, and used as a model mesophase pitch precursor in this study without further treatment. ARMP has a softening point at 240 °C and carbon aromaticity of 0.84 and 100 vol. %, in terms of the mesophase texture. Table A1 (in Appendix A) summarizes its general properties [34,35].

### 2.2. Spinning of Mesophase Pitch

ARMP was melt spun into as-spun AR mesophase pitch fiber (ARMP-F) using a single-hole spinneret at 340 °C with a laboratory-type mono-hole melt spinning apparatus, which, as a spinneret, has a stainless-steel die hole with a diameter and length of 0.3 and 0.6 mm (L/D = 2), respectively [35,36]. Figure 1 shows a schematic diagram of the self-designed laboratory-type spinning apparatus. The spinning conditions were carefully controlled to diameters of spun mesophase pitch fibers of just 10.5 ± 1.0 and 14.0 ± 1.0 µm, which are designated as ARMP-F10 and ARMP-F14, respectively. ARMP-F14 was used to examine the effect of pressure on the oxygen diffusivity and the oxidation reactions in stabilization, and ARMP-F10 was used to prepare the carbonized and graphitized fibers (ARMP-CF10 and ARMP-GF10) to evaluate the mechanical performance.

### 2.3. Oxidation–Stabilization of Spun Mesophase Fibers

Oxidation–stabilization of spun mesophase pitch fibers was carried out through thermal oxidation under dried air flow of 0.1–1.0 MPa with a flow rate of 100 mL/min. Figure 2 shows a schematic diagram and an actual image of the custom-designed oxidation–stabilization apparatus using atmospheric and pressurized air flows. Oxidation–stabilization of ARMP-F14 was carried out at 270 °C with a heating rate of 2.0 °C/min for 20 min under air flow pressures of 0.1, 0.5, and 1.0 MPa to evaluate the oxygen distribution across the fiber cross section. Oxidation–stabilization of ARMP-F10 was carried out at various temperatures, heating rates, soaking times, and air flow pressures of 250–270 °C, 0.5–3.0 °C/min, 0–60 min, and 0.1–1.0 MPa, respectively, to monitor the oxygen uptake and prepare the optimally oxidative –stabilized fibers for evaluating the mechanical performances of carbonized and graphitized fibers under each stabilization condition. The stabilized fibers of ARMP-F10 and ARMP-F14 are designated as ARMP-SF10 and ARMP-SF14, respectively.

### 2.4. Carbonization and Graphitization of Stabilized Fiber

After stabilization of the ARMP-Fs under various conditions, ARMP-SF14s were successively heat-treated at 800 °C for 5 min with a heating rate of 5.0 °C/min in a nitrogen atmosphere for carbonization, and further heat-treated at 2400 °C for 10 min with a heating rate of 15 °C/min under an argon atmosphere for graphitization. ARMP-SF10s were heat-treated at 1000 °C for 30 min with a heating rate of 20 °C/min in a vacuum atmosphere, and some of the carbonized fibers were also further graphitized at 2800 °C for 10 min with a heating rate of 15 °C/min in an Ar atmosphere to evaluate the mechanical performance. The carbonized and graphitized fibers of ARMP-SF10 and ARMP-SF14 are designated ARMP-CF10 and ARMP-CF14, and ARMP-GF10 and ARMP-GF14, respectively.

### 2.5. Characterisation

Spun mesophase pitch fibers were subjected to TGA to track the amount of oxygen uptake under atmospheric and air flow pressure conditions using a magnetic suspension balance (MicrotracBEL MSB-TG-1300; Belsorp Co. Ltd., Osaka, Japan). Figure A1 shows a schematic of MSB-TG-1300. TGA was carried out under various heating rates and air flow pressures of 1.0–4.0 °C/min and 0.1–1.0 MPa, respectively. An exclusive alumina pan (diameter: 10 mm, height: 10 mm, weight: 6.11 g) was used under the controlled flows of air/oxygen with a flow rate of 100 mL/min. After TGA, the activation energy of the oxidation–stabilization of ARMP-F14s was calculated using Equation (1) of Kissinger’s method [37]: ln (*q*/*T*_max_^2^) = −*E*_a_/(*RT*_max_),(1)
where *T*_max_, *q*, and *E*_a_ designate peak temperature, heating rate, and activation energy, respectively.

To evaluate the distribution of the amounts of oxygen uptake in the transverse sections of stabilized fibers, ARMP-SF14s were analyzed using a scanning electron microscope (SEM) equipped with an electron probe micro-analyzer (EPMA) (JSM-6340F; JEOL, Tokyo, Japan) [38]. Images of the structure of the transverse sections of the graphitized fibers were obtained using a scanning electron microscope (JSM-6700F; JEOL, Tokyo, Japan). The surface morphology and mean diameter of the resulting carbon fibers were also measured. 

Elemental analysis was used to determine the total amount of oxygen uptake of the stabilized fibers for evaluating the mechanical performance of the obtained carbonized and graphitized fibers. The mechanical properties of the carbonized and graphitized fibers were measured using a universal tensile tester (TENSILON/UTM-II-20; ORIENTEC, Tokyo, Japan), in accordance with the JIS R 7606:2000 method (a method of single filament test) [39]. Twenty-five filaments were tested for obtaining the averaged mechanical properties. The averaged diameters of carbonized and graphitized fibers were checked with SEM (JSM6700, JEOL, Tokyo, Japan) measurements with 5 kV of acceleration voltage.

## 3. Results and Discussion

### 3.1. Stabilization of Mesophase Pitch Fibers under Atmospheric and Pressurised Conditions

Figure 3 shows the TGA profiles of oxygen uptakes of ARMP-F14 and ARMP-F10 in oxidation–stabilization with a heating rate of 2.0 °C/min under atmospheric and pressurized air flow conditions. ARMP-F10 and ARMP-F14 had very similar profiles of oxygen uptake, which means that they experienced almost the same oxidation reactions under the same applied pressure. Figure 4 shows the TGA profiles of the oxygen uptakes of ARMP-F14 under the various heating rates and air flow pressures. The results clearly reveal two interesting distinctions, namely, in the temperatures of the starting and maximum oxidations, and the amounts of oxygen uptake. First, the oxidation reaction for oxygen uptake occurred earlier in the stabilized fibers under air flow pressures of 0.5 and 1.0 MPa than for the stabilized one under atmospheric pressure. The start of oxygen uptake of ARMP-F14 mainly occurred at over 150 °C under atmospheric pressure regardless of the heating rate, whereas the pressurized stabilization under 0.5 and 1.0 MPa allowed this to occur at around 125 °C. During the manufacture of mesophase pitch-based carbon fibers, the reactions that typically occur during the stabilization process are oxidation, dehydration, condensation, oxidation crosslinking, elimination of volatile matter, and oxidative decomposition [40]. Such complexity in oxidation–stabilization reactions makes it difficult to obtain a comprehensive understanding of the stabilization of mesophase pitch materials. However, Yoon et al. successfully optimized the stabilization process by performing simple monitoring of oxygen uptake using TGA with several heating rates [26]. For TGA analysis, it allowed only two reactions of the oxidation and oxidative decomposition of alkyl and aromatic molecules as main reactions and proved that the oxidative decomposition of alkyl groups always occurred before that of condensed aromatic ones. On the basis of this analysis, they explained that oxygen uptake occurred at a lower temperature and higher rate with a decrease of the heating rate in oxidation–stabilization. They proved that a higher heating rate, which inevitably required a higher temperature to complete the optimal stabilization, easily incurred excess oxidation in the outer section of mesophase phase pitch fibers, and such excess oxidation could trigger decomposition of the alkyl groups of pitch molecules with the dissipation of decomposed gases such as CO and CO_2_ before the main decomposition of aromatic molecules of mesophase pitch fibers. In these results, the temperature at which the maximum oxygen uptake in oxidation–stabilization occurred was shifted to a lower position with increasing air pressure. This proved that the main oxygen uptake could start at a lower temperature under pressurized air flow conditions than under atmospheric conditions, and the application of pressure in oxidation–stabilization had a similar effect to decreasing the heating rate. Such an earlier start of oxidation and higher amounts of oxygen uptake are well matched with the previous reports [30,31].

Table 1 summarizes the activation energies of the oxidation reactions in various stabilized conditions from the calculation based on Equation (1) using Arrhenius plots shown in Figure 5. The activation energies at 0.5 and 1.0 MPa were 230 and 271 kJ/mol, respectively, which was less than almost half the value of 535 kJ/mol at atmospheric pressure. 

Regarding the maximum amount of oxygen uptake in the oxidation–stabilization with applied pressure, different effects of decreasing the heating rate were observed. Specifically, the stabilized fiber of ARMP-SF14 at 0.5 MPa had a higher amount of oxygen uptake at its maximum point of oxygen uptake than the stabilized fiber under atmospheric conditions. However, the maximum amount of oxygen uptake was higher at 0.5 MPa, but slightly lower at 1.0 MPa compared with those under atmospheric pressure conditions. As is clearly shown in Figure 4 and Figure 5, the oxidation rate under the pressure conditions of 1.0 MPa was reliably higher than that under atmospheric conditions. To understand these results, we must consider the oxygen diffusivity and oxidation reaction in combination.

Figure 6 shows the SEM-EPMA (scanning electron microscope (SEM) equipped with an electron probe micro-analyzer) results of ARMP-SF14s stabilized under air flow pressures of 0.1, 0.5, and 1.0 MPa with a heating rate of 2.0 °C/min. The distribution of oxygen uptakes shifted to a higher level and became more homogeneous with the increasing applied pressure of air flow, with the oxygen distribution in ARMP-SF under pressure of 1.0 MPa ranging from 12.1 to 14.8 wt. %. The distribution of the amounts of oxygen uptake appeared relatively homogeneous in ARMP-SF14 stabilized at an air pressure of 0.5 MPa, with the amounts of oxygen uptake ranging from 8.7 to 11.4 wt. %. Compared with the distribution of the amounts of oxygen uptake of ARMP-SF14 stabilized under pressurized air flow conditions, that of ARMP-SF14 stabilized under atmospheric conditions was very heterogeneous, with the amounts of oxygen uptake of 3.4 to 11.2 wt. %. From these results, we can see that the oxygen diffusion from the outer surface to the center part of the pitch fiber becomes more rapid with increasing applied air flow pressure; coincidently, the oxidation reaction is also more rapid because the oxygen density is higher under the pressure conditions. That is, in the more rapid and homogeneous oxidation reactions that occurred at a pressure of 1.0 MPa, oxidation can easily occur, but an excess oxidation reaction may also occur, which would be a clear reason for the weight loss associated with attaining the maximum oxygen uptake position of the mesophase pitch fibers.

Figure 7 shows SEM images of the transverse sections of graphitized fibers stabilized under air pressures of 0.1, 0.5, and 1.0 MPa. ARMP-GF14 in Figure 7a shows the typical skin-core structure of mesophase pitch-based graphitized fibers [41,42], but this is not the case for ARMP-GF14s in Figure 7b,c. Mochida et al. reported that a high heating rate and low final temperature in stabilization are responsible for the formation of a distinct skin-core structure, which is one of the main factors lowering the tensile strength of the obtained graphitized fibers [25]. Here, the rapid heating and low final temperature in the stabilization conditions resulted in stabilized fibers with a deficient oxygen reaction in their center. Such a deficiency in the amounts of oxygen uptake in the thermal oxidation–stabilization is the main reason for the formation of the skin-core structure. Our results, shown in Figure 6 and Figure 7, confirm the fact that oxygen uptake of at least more than 7.0 wt. % in stabilized mesophase fibers is necessary under the present stabilization conditions to obtain carbonized and graphitized fibers that do not have the skin-core structure. The upper limit of the optimized oxygen uptake is still difficult to determine. The stabilized fibers that allow the highest yields after carbonization and graphitization and the highest mechanical performance, especially highest tensile strength, should have the most optimized oxidation state. The upper limit of oxygen uptake must be determined with such optimized stabilized fibers. From the above oxygen uptake criteria, ARMP-SF stabilized under air pressure of 0.5 MPa may approach the most stabilized state, whereas ARMP-SF stabilized under air pressures of the atmospheric level and 1.0 MPa of the pressurized level show different oxygen uptakes of the deficient and excess oxygen weight gains in the center and outer parts of pitch fibers, respectively.

### 3.2. Oxidation–Stabilization of Mesophase Pitch Fiber Using Laboratory Stabilization Apparatus

Table 2 and Table A2 (in Appendix A) show the results of oxidation–stabilization of ARMP-F10s under various pressurized air flows. In the oxidation–stabilization using laboratory-type stabilization apparatus, the maximum stabilization temperature was limited to 270 °C to achieve oxidation reactions that were as excessive as possible, which should exhibit some differences in the oxidation state of stabilized fibers in the TGA analyses.

In the oxidation–stabilization under atmospheric conditions, the amounts of oxygen uptakes of the stabilized fibers with a soaking time of 0 min at temperatures of 250, 260, and 270 °C with a heating rate of 2.0 °C/min were evaluated as being 5.2, 6.2 and 8.5 wt. %, respectively. The amounts of oxygen uptakes with a heating rate of 3.0 °C/min after soaking for 0, 30, and 60 min at 270 °C were 6.5, 7.8 and 8.5 wt. %, respectively. The increase of the amounts of oxygen uptake for soaking times between 30 and 60 min was very small because the ARMP-F10 almost reached a fully oxidized state under atmospheric air flow pressure with a soaking time of 60 min at 270 °C and a heating rate of 3.0 °C/min. Compared with this, the amounts of oxygen uptake with a heating rate of 0.5 °C/min after soaking for 0 min at 270 °C was higher with 11.7 wt. %. This higher amount of oxygen uptake at 0.5 °C/min demonstrates that a more stabilized state of ARMP-F10 could be obtained with a decreased heating rate of 0.5 °C/min in oxidation–stabilization under atmospheric air flow pressure.

In the oxidation–stabilization under air flow pressure of 0.5 MPa, the amounts of oxygen uptake of the stabilized fibers with a soaking time of 0 min at temperatures of 250, 260, and 270 °C and a heating rate of 2.0 °C/min were 6.3, 11.4 and 11.9 wt. %, respectively. The amount of oxygen uptake after soaking for 15 min at 260 °C with a heating rate of 3.0 °C/min was 11.1 wt. % and those with soaking times of 0 and 10 min at 270 °C with the same heating rate were 10.8 and 11.8 wt. %, respectively. The increase of oxygen uptake between 260 and 270 °C with 0 min of soaking was very small and the oxygen uptake after soaking for 10 min at 270 °C with a heating rate of 3.0 °C/min was also 11.8 wt. %, which indicates that ARMP-F10 was almost fully stabilized upon soaking for 10 min at 270 °C with a heating rate of 3.0 °C/min under air flow pressure of 0.5 MPa.

In the oxidation–stabilization under air flow pressure of 1.0 MPa, the amounts of oxygen uptakes of the stabilized fibers at soaking temperatures of 250, 260, and 270 °C with a heating rate of 2.0 °C/min were 7.0, 11.5 and 11.2 wt. %, respectively. The amount of oxygen uptake after soaking for 0 min at 260 °C with a heating rate of 3.0 °C/min was 11.5 wt. %, and those after soaking for 0 and 5 min at 270 °C with the same heating rate were 11.2 and 11.1 wt. %, respectively. The amounts of oxygen uptakes of the stabilized fiber soaked at 270 °C were slightly lower than those of the stabilized fiber at 260 °C, which indicated that the oxidation decomposition occurred at 270 °C with a heating rate of 2.0 °C/min through excess oxidation reactions of mainly the exterior of the stabilized fiber. The decrease in oxygen uptakes with soaking times of 0 and 5 min at 270 °C with a heating rate of 3.0 °C/min were also the result of oxidation decomposition. The amount of oxygen uptake after soaking for 0 min at 270 °C with a heating rate of 3.0 °C/min was 11.2 wt. %. From these results, ARMP-SF10 was fully or excessively stabilized with soaking for 0 min at 270 °C with a heating rate of 3.0 °C/min under air flow pressure of 1.0 MPa.

### 3.3. Yields of Carbonization and Graphitization of the Stabilized Fibers and the Mechanical Performances of the Carbonized and Graphitized Fibers

Table 2 and Table A2 (in Appendix A), Table 3 and Table A3 (in Appendix A) also list the yields of carbonization and graphitization of the oxidative –stabilized fibers and the mechanical performances of carbonized and graphitized fibers. The yields of carbonization and graphitization were evaluated by the weight ratios (percentages) of the carbonized and graphitized fibers relative to the mesophase pitch fibers.

The carbonization yields of ARMP-CFs stabilized with soaking for 0 min at 250, 260, and 270 °C with a heating rate of 2.0 °C/min under atmospheric pressure were 81.0, 85.2, and 83.2 wt. %, respectively. Moreover, the yields of ARMP-CFs stabilized with soaking for 0, 30, and 60 min at 270 °C with a heating rate of 3.0 °C/min under atmospheric pressure were 83.2, 85.6 and 87.0 wt. %, respectively. The carbonization yield of ARMP-CF stabilized with soaking for 0 min at 270 °C with a heating rate of 0.5 °C/min under atmospheric pressure was 88.3 wt. %. Furthermore, the carbonization yields of ARMP-CFs stabilized with soaking for 0 min at 250, 260, and 270 °C with a heating rate of 2.0 °C/min under air flow pressure of 0.5 MPa were 90.0, 90.9, and 89.1 wt. %, respectively. The yields of ARMP-CFs stabilized with soaking for 15 min at 260 °C and soaking for 0 and 10 min at 270 °C with a heating rate of 3.0 °C/min under air flow pressure of 0.5 MPa were 88.8, 89.0, and 89.5 wt. %, respectively. The analyses also revealed that the carbonization yields of ARMP-CFs stabilized with soaking for 0 min at 250, 260, and 270 °C with a heating rate of 2.0 °C/min under air flow pressure of 1.0 MPa were 85.3, 85.4, and 85.3 wt. %, respectively. The yields of ARMP-CFs stabilized with soaking for 0 min at 260 °C and soaking for 0 and 5 min at 270 °C with a heating rate of 3.0 °C/min under air flow pressure of 1.0 MPa were 89.6, 85.2, and 84.3 wt. %, respectively. From these results, if the stabilization under air flow pressures of 0.5 and 1.0 MPa could be carried out on optimal conditions (soaking time and temperature), the carbonized fibers that were stabilized under air flow pressure of 0.5 and 1.0 MPa had higher carbonization yields than those stabilized under air flow pressures of the atmospheric level. Generally, the stabilized fibers with oxygen uptakes of less than 7.0 wt. % and more than 13.0 wt. % had lower carbonization yields than the stabilized fibers with oxygen uptakes of around 11.0–12.0 wt. %. These results suggest that oxygen uptakes of 11.0–12.0 wt. % might be close to the optimal oxidation state for the oxidation–stabilization of ARMP-F10. The deficiency of oxygen uptake in the fibers stabilized at 270 °C with heating rates of 2.0 and 3.0 °C/min under the atmospheric air flow pressure was ascribed to the insufficient delivery of oxygen into the center of the pitch fibers under these conditions, conferring excess volatility of light aromatic molecules in the carbonization. In contrast, excessive oxygen uptake in the stabilized fibers usually occurred under an air flow pressure of 1.0 MPa in our study, indicating that the excess oxidation mainly occurred in the exterior of pitch fibers, which might be the principal reason for the oxidation decomposition of pitch and stabilized fibers in the stabilization and carbonization processes.

The graphitization yields revealed a trend very similar to those of carbonization. The average weight loss in the graphitization from the carbonization was estimated to be approximately 1.0–2.0 wt. %. The graphitized fibers stabilized under an air flow pressure of 0.5 MPa had higher graphitization yields than those stabilized under air flow pressure conditions of the atmospheric level and 1.0 MPa.

With regards to the mechanical performance of the carbonized and graphitized fibers stabilized with soaking for 0 min at 270 °C with a heating rate of 2.0 °C/min under atmospheric pressure, they had tensile strength of 2.4 and 3.5 GPa, elongation ratios of 1.5% and 0.6%, and Young′s moduli of 159 and 508 GPa, respectively. The carbonized and graphitized fibers stabilized with soaking for 0 min at 270 °C with a heating rate of 0.5 °C/min under atmospheric pressure had markedly improved values of tensile strengths of 2.9 and 4.0 GPa, elongation ratios of 1.7% and 0.6%, and Young′s moduli of 171 and 663 GPa, respectively. The carbonized and graphitized fibers stabilized with soaking for 0 min at 260 °C with a heating rate of 2.0 °C/min under air flow pressure of 0.5 MPa had the best mechanical performance of tensile strengths of 3.4 and 4.6 GPa, elongation ratios of 1.7% and 0.6%, and Young′s moduli of 177 and 765 GPa, respectively. These results indicate that tensile strength, elongation ratio, and Young’s modulus of the carbonized fibers of higher than 1.7 GPa, 1.7%, and 170 GPa, respectively, after carbonization at 1000 °C for 30 min, were successfully obtained within a total stabilization time of less than 60 min. From these results, stabilization under air pressures of 0.5 MPa cause homogeneous oxidation and increases mechanical properties more than other graphitized fibers. In contrast, the mechanical properties of ARMP-GF10 stabilized under air pressures of the atmospheric level and 1.0 MPa may decrease owing to the deficient and excess oxygen uptake in the center and outer parts of pitch fibers, respectively.

Figure 8 schematically shows the oxidation and oxidation decomposition in the oxidation–stabilization at 270 °C for soaking for 0 min with a heating rate of 2.0 °C/min under air flow pressures at the atmospheric level, 0.5, and 1.0 MPa. From the model images, we can assume that oxidation–stabilization occurs via the following mechanism, as summarized in Table 4. Specifically, appropriate oxidation reactions can be achieved for the optimization of pitch fibers through oxidation–stabilization under a mild air flow pressure of 0.5 MPa with a relatively short total stabilization time.

## 4. Conclusions

It usually takes more than 4 h to carry out the optimal oxidation–stabilization of mesophase pitch fiber to obtain the maximum mechanical performances, especially the tensile strength. The oxidation–stabilization under mild air flow pressures of 0.5 and 1.0 MPa could greatly shorten the total stabilization time to less than 60 min to obtain mesophase pitch-based carbonized and graphitized fibers without deteriorating the mechanical performance. Notably, carbonized fibers with high tensile strengths and Young′s moduli of over 3.0 and 170 GPa, respectively, which had only been heat-treated at 1000 °C for 30 min, were successfully obtained with a total stabilization time of less than 60 min through oxidation–stabilization under air flow pressures of 0.5 MPa.

Activation energies for oxidation reactions in stabilization under air flow pressure are much lower than those of oxidation reactions under atmospheric air flow pressure because of the higher diffusivity of oxygen into the center and a more rapid oxidation reaction on the molecules of mesophase pitch fibers under mild air flow pressures of 0.5 and 1.0 MPa. The higher oxygen diffusivities resulted in a more homogeneous distribution of oxygen uptake across the transverse section of mesophase pitch fibers and allowed higher yields of carbonization and graphitization, which are directly related to the improvement of the mechanical properties. Additionally, excess oxidation can bring about the oxidation decomposition of pitch molecules with oxidation–stabilization under air flow pressures of 0.1 and 1.0 MPa, which decreases the carbonization yield and tensile strength of the obtained carbon fibers.

## Figures and Tables

**Figure 1 polymers-11-01911-f001:**
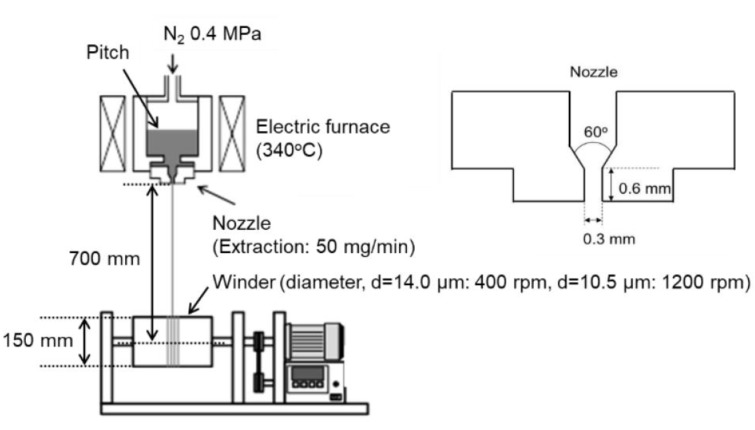
Schematic picture of the self-designed laboratory-type mono-hole melt spinning apparatus.

**Figure 2 polymers-11-01911-f002:**
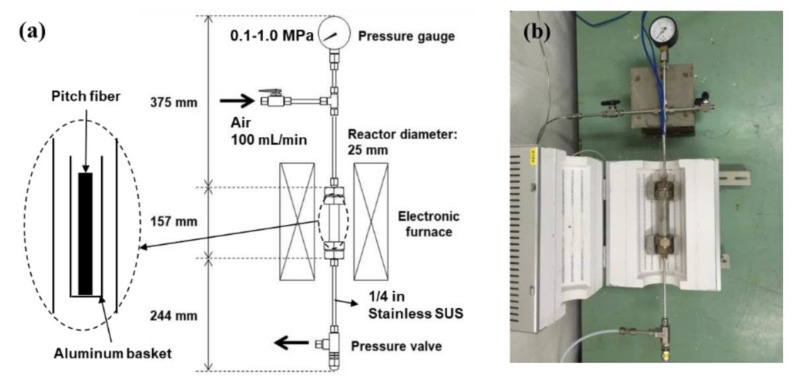
The schematic and real pictures of self-designed laboratory-type oxidative stabilization apparatus: (**a**) schematic picture of self-designed laboratory-type oxidative stabilization apparatus and (**b**) real picture of self-designed laboratory-type oxidative stabilization apparatus.

**Figure 3 polymers-11-01911-f003:**
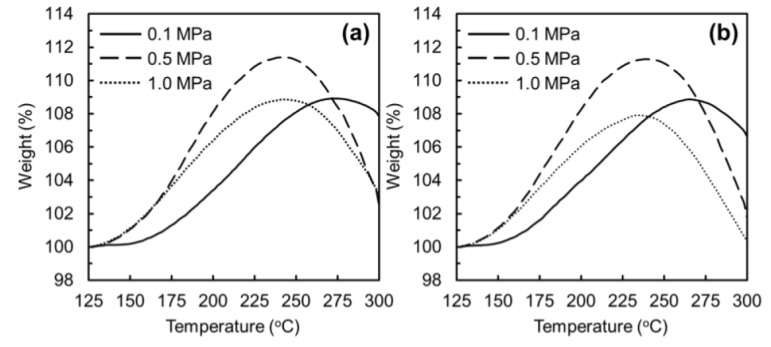
Thermo-gravimetric analysis (TGA) profiles of the oxidation–stabilization of mesophase pitch fibers with different applied pressures: (**a**) AR mesophase pitch fiber (ARMP-F)14 and (**b**) ARMP-F10.

**Figure 4 polymers-11-01911-f004:**
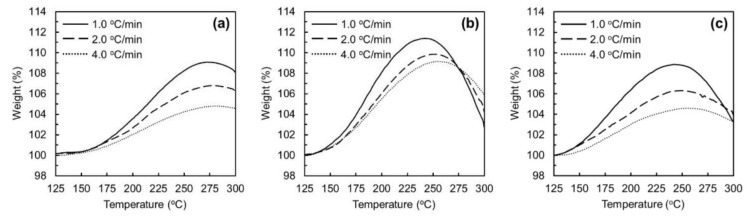
TGA profiles of oxygen uptakes in the oxidation–stabilizations of mesophase pitch fibers with different heating rates: (**a**) 0.1 MPa, (**b**) 0.5 MPa, and (**c**) 1.0 MPa.

**Figure 5 polymers-11-01911-f005:**
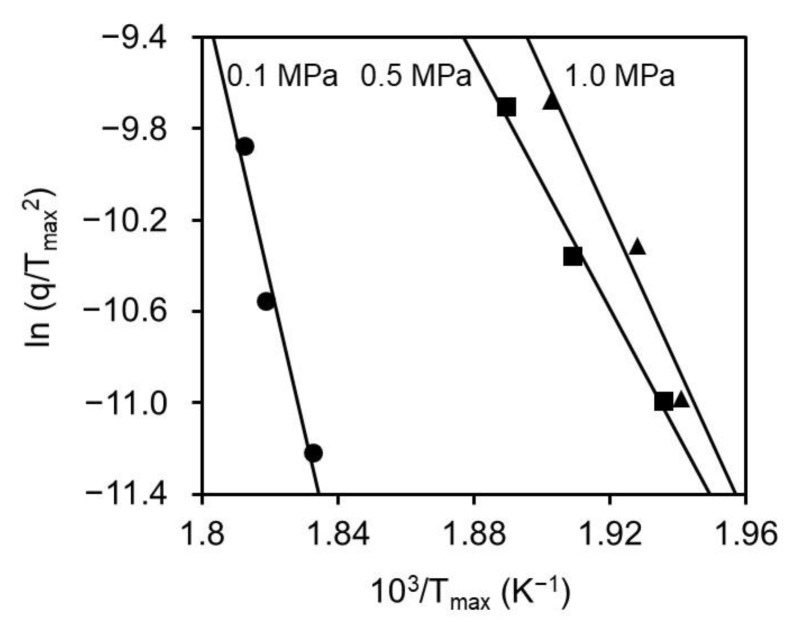
Arrhenius plots of the oxidation–stabilization of mesophase pitch fibers with different applied pressure.

**Figure 6 polymers-11-01911-f006:**
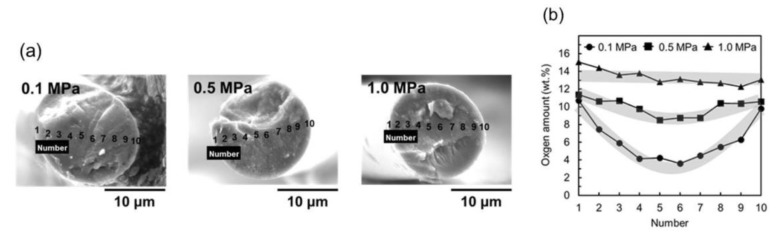
Distributions of the amounts of oxygen uptakes in the transversal sections of ARMP stabilized fiber (ARMP-SF)14s using SEM-EPMA: (**a**) measuring points of the amounts of oxygen uptakes in ARMP-SF14s and (**b**) distributions of the amounts of oxygen uptakes in the transversal sections of ARMP-SF14s.

**Figure 7 polymers-11-01911-f007:**
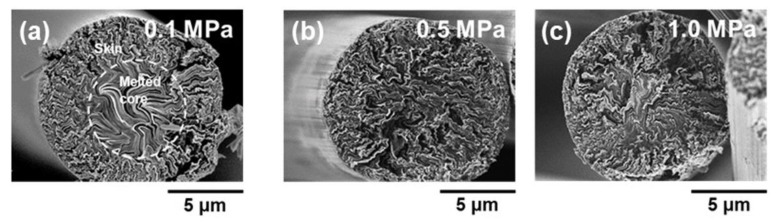
SEM images of the transverse sections of graphitized fibers stabilized under various air flow pressures: (**a**) 0.1 MPa, (**b**) 0.5 MPa, and (**c**) 1.0 MPa.

**Figure 8 polymers-11-01911-f008:**
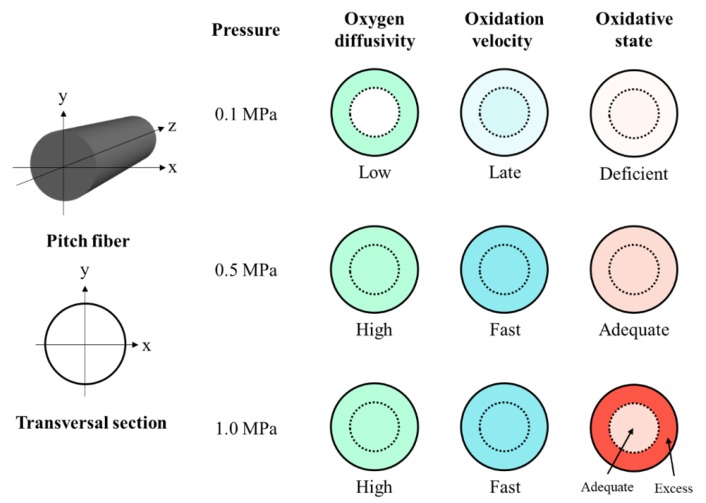
Schematic model pictures of the oxidation and oxidation decomposition in the oxidation–stabilization at 270 °C for soaking for 0 min with a heating rate of 2.0 °C/min under air flow pressures at the atmospheric level (0.1, 0.5, and 1.0 MPa).

**Table 1 polymers-11-01911-t001:** Activation energies on the oxidation reactions of mesophase pitch under various applied pressures.

Applied Pressure (MPa)	0.1	0.5	1.0
Activation Energy (kJ/mol)	535	230	271

Condition of oxidation reactions of mesophase pitch under various applied pressure; heating rates: 1.0, 2.0, and 4.0°C/min; applied pressure: 0.1, 0.5, and 1.0 MPa, with pressurized air flow of 100 mL/min.

**Table 2 polymers-11-01911-t002:** The results of the stabilization under atmospheric and pressurized conditions using laboratory stabilization apparatus, yields, and mechanical properties of carbonized fibers (heating rate of stabilization: 2.0 and 3.0 °C/min, soaking time: 0–60 min). ARMP-CF, AR mesophase pitch carbonized fiber.

Stabilization Condition of Mesophase Pitch Fiber (ARMP-F10)						Yield and Mechanical Properties of Carbonized Fiber (ARMP-CF10)^*3^				
Applied Pressure	Heating Rate	Soaking Temperature	Soaking Time	Oxygen Uptake^*1^	Total Time^*2^	Yield	Diameter	Tensile Strength	Elongation Ratio	Young′s Modulus
(MPa)	(°C/min)	(°C)	(min)	(wt. %)	(min)	(wt. %)	(μm)	(GPa)	(%)	(GPa)
0.1	2.0	270	0	8.5	60	83.2	7.8 ± 0.2	2.4 ± 0.3	1.5	159 ± 27
	3.0		60	8.5	100	87.0	7.7 ± 0.2	2.4 ± 0.4	1.5	142 ± 34
0.5	2.0	270	0	11.9	60	89.1	7.7 ± 0.2	2.8 ± 0.3	1.8	186 ± 28
	3.0		10	11.8	50	89.5	7.8 ± 0.1	2.4 ± 0.3	1.7	191 ± 31
1.0	2.0	270	0	11.2	60	85.3	7.5 ± 0.1	3.2 ± 0.3	1.5	161 ± 48
	3.0		5	11.1	45	84.3	7.5 ± 0.3	2.6 ± 0.5	1.6	159 ± 39

*1: Oxygen up-take was determined by elemental analysis; *2: total time was determined by the summing of heating time from 150 °C to soaking temperature and soaking time at soaking temperature; *3: carbonization was carried out at 1000 °C for 30 min with the heating rate of 15 °C/min under vacuum.

**Table 3 polymers-11-01911-t003:** The yields and mechanical properties of graphitized fibers (Heating rate of stabilization: 2.0 and 3.0 °C/min, soaking time: 0–60 min). ARMP-GF, ARMP graphitized fiber.

Stabilization Condition of Mesophase Pitch Fiber (ARMP-F10)				Yield and Mechanical Properties of Graphitized Fiber (ARMP-GF10)^*1^				
Applied Pressure	Heating Rate	Soaking Temperature	Soaking Time	Yield	Diameter	Tensile Strength	Elongation Ratio	Young’s Modulus
(MPa)	(°C/min)	(°C)	(min)	(wt. %)	(μm)	(GPa)	(%)	(GPa)
0.1	2.0	270	0	82.4	7.7 ± 0.1	3.5 ± 0.3	0.6	508 ± 30
	3.0		60	85.2	7.5 ± 0.1	3.3 ± 0.3	0.5	666 ± 50
0.5	2.0	270	0	86.8	7.6 ± 0.2	4.6 ± 0.3	0.6	767 ± 39
	3.0		10	86.2	7.6 ± 0.1	4.4 ± 0.3	0.6	740 ± 43
1.0	2.0	270	0	83.0	7.4 ± 0.1	3.5 ± 0.2	0.5	725 ± 41
	3.0		5	82.9	7.3± 0.1	3.9 ± 0.5	0.6	667 ± 55

*1: Graphitization was carried out at 2800 °C for 10 min with the heating rate of 20 °C under an Ar atmosphere.

**Table 4 polymers-11-01911-t004:** Estimated oxidation state of the stabilized fiber under various air flow pressures based on the model mechanisms of oxidation diffusion and reaction.

Stabilization Conditions*				Oxygen Diffusivity From Outer Surface to Center Part of Pitch Fiber	Oxidation Reaction and Oxidized State on Molecules of Mesophase Pitch Fiber	
Applied Pressure (MPa)	Heating Rate (°C/min)	Soaking Temperature (°C )	Soaking Time (min)		Outer Part of Fiber	Center Part of Fiber
0.1	2.0	270	0	Low	Late, deficient	Late, deficient
	3.0		60	Low	Late, excess	Late, deficient
0.5	2.0	270	0	High	Fast, adequate	Fast, adequate
	3.0		10	High	Fast, adequate	Fast, adequate
1.0	2.0	270	0	High	Fast, excess	Fast, adequate
	3.0		5	High	Fast, excess	Fast, excess

* Amount of air flow: 100 mL/min.

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
