# Peer review of "Shortening Stabilization Time Using Pressurized Air Flow in Manufacturing Mesophase Pitch-Based Carbon Fiber"

_polymers, 2019, doi:10.3390/polym11121911_

Round 1
Reviewer 1 Report
The article can be published in the present form.
Author Response
Thank you very much for your kind comments.
Reviewer 2 Report
For pitch based carbon fiber, the stabilization is time consuming, especially for meso-phase pitch. Actually, there are few studies on pressurized air flow, since this method is not expected to be practically adopted in a continuous line at a reasonable cost. In fact, there are also few studies on the stabilization in high oxygen content air. Under certain condition, the pressurized air could be considered as high oxygen content air at normal pressure. Although the proposed method could not be practically adopted in industry, these studies would be useful for better understanding the role of oxygen diffusion on the structures and performances of the resulting carbon fibers. Thus, I would recommend the possible publication of the manuscript in POLYMERS after proper revision according to the following comments. 1. Line 308: “… whereas ARMP-SFs stabilized under air pressures of the atmospheric level and 1.0 MPa have excess and deficient oxygen uptake in the outer and center parts of pitch fibers, respectively.” Difficult to catch the meaning of this sentence. 2. Tables 3 and 4: In my opinion, when air pressure increases from 0.5MPa to 1.0MPa, the required stabilization temperature and durations would be less. It could be found that the carbonized fibers at a heating rate of 3C/min, a temperature of 260C, and an air pressure of 1MPa possess the highest carbon yield and modulus among all fibers. Thus, the increase of air pressure from 0.5 MPa to 1 MPa would not cause excess oxidation, and one has to optimize the temperature and duration according to the air pressure. Please check and revise the descriptions in line 378 and other places: “From these results, the carbonized fibers stabilized under air flow pressure of 0.5 MPa had higher carbonization yields than those stabilized under air flow pressures at the atmospheric level and 1.0 MPa.” I think that you cannot simply state that the stabilization at 0.5 MPa is better than that at 1 MPa. 3. Figure 7c: Beside oxygen content distribution in the stabilized fibers, could author kindly provide similar Raman data on the carbonized fibers? And, it will be easier to under the correlation between stabilized structures and the resultant carbonized structures. 4. Tables 3 and 4: I am wondering that whether the modulus values have been calibrated by instrument compliance?Author Response
Dear reviewers:
Thank you for your letter and the reviewers’ comments concerning our manuscript. Those comments are very much valuable and helpful for revising and improving our paper, as well as important for our further research. We have studied the comments carefully and have made the corrections. The responds to the reviewers’ comments are listed as follows:
Our response (in regular type) to the comments (in italics) follows:
Line 308: “… whereas ARMP-SFs stabilized under air pressures of the atmospheric level and 1.0 MPa have excess and deficient oxygen uptake in the outer and center parts of pitch fibers, respectively.” Difficult to catch the meaning of this sentence.
Answer:
Thank you very much for kind comment.
Line 308 (Line 298) was revised. “whereas ARMP-SF stabilized under air pressures of the atmospheric level and 1.0 MPa of the pressurized level show the different oxygen uptakes of the deficient and excess oxygen weight gains in the center and outer parts of pitch fibers, respectively.”
Tables 3 and 4: In my opinion, when air pressure increases from 0.5MPa to 1.0MPa, the required stabilization temperature and durations would be less. It could be found that the carbonized fibers at a heating rate of 3C/min, a temperature of 260C, and an air pressure of 1MPa possess the highest carbon yield and modulus among all fibers. Thus, the increase of air pressure from 0.5 MPa to 1 MPa would not cause excess oxidation, and one has to optimize the temperature and duration according to the air pressure. Please check and revise the descriptions in line 378 and other places: “From these results, the carbonized fibers stabilized under air flow pressure of 0.5 MPa had higher carbonization yields than those stabilized under air flow pressures at the atmospheric level and 1.0 MPa.” I think that you cannot simply state that the stabilization at 0.5 MPa is better than that at 1 MPa.
Answer:
Thank you for kind comment. The reviewer’s comment is right.
Line 372 (Line 368) was revised. “From these results, if the stabilization under air flow pressures of 0.5 and 1.0 MPa could be carried out on optimal conditions (soaking time and temperature), the carbonized fibers that were stabilized under air flow pressure of 0.5 and 1.0 MPa had higher carbonization yields than those stabilized under air flow pressures of the atmospheric level.”
Figure 7c: Beside oxygen content distribution in the stabilized fibers, could author kindly provide similar Raman data on the carbonized fibers? And, it will be easier to under the correlation between stabilized structures and the resultant carbonized structures.
Answer:
Thank you for kind comment. According to reviewer’s comment, Raman spectroscopy might allow us a valuable data. However, Raman spectroscopy is usually very sensitive to surface but not bulk, and we only obtained the supported graphitic properties in carbonized and graphitized fibers. From these reasons, we did not try to obtain the Raman data. Next time, we will try to obtain the Raman data on the carbonized fibers and compared them with SEM cross-sectional textures.
Tables 3 and 4: I am wondering that whether the modulus values have been calibrated by instrument compliance?
Answer: Yes, the modulus values have been calibrated by instrument compliance.
Reviewer 3 Report
The subject matter of the manuscript is relevant for publication in the journal and this well-written manuscript can be considered for publication after revision. I would ask that the authors to consider following recommendations for a minor revision:
“Table 1 summarizes its general properties [33,34]” – Please consider moving the data to supplementary information if not measured by the authors. “ARMP-F14 was used to examine the effect of pressure on the oxygen diffusivity and the oxidation reactions in stabilization, and ARMP-F10 was used to prepare the carbonized and graphitized fibers” – Why didn’t the authors use the same diameter for the both? “Figure 3 shows a schematic of MSB-TG-1300.” –Please consider moving the schematic to supplementary information. “ARMP-F10 and ARMP-F14 had very similar profiles of oxygen uptake, which means that they experienced almost the same oxidation reactions under the same applied pressure.” – Did the authors expect any significant change in oxygen uptake since the diameter difference between these two samples was very close? Could you please explain why such close fiber diameter difference was selected for this study? Table 3 and 4 provide a lot of data, however, it is hard to follow any trend. I am recommending obtaining some graphs out of these tables showing the significant trends only and to move the tables in the supplementary information for further reader’s interest. Page 12 (395-407): Here authors provide observations on the mechanical performance of the fibers without much insight or explanation. Please explain the trends as can be seen in the tables including the mechanism.
Author Response
Dear reviewers:
Thank you for your letter and the reviewers’ comments concerning our manuscript. Those comments are very much valuable and helpful for revising and improving our paper, as well as important for our further research. We have studied the comments carefully and have made the corrections. The responds to the reviewers’ comments are listed as follows:
Our response (in regular type) to the comments (in italics) follows:
Reviewer 3
“Table 1 summarizes its general properties [33,34]” – Please consider moving the data to supplementary information if not measured by the authors.
Answer:
Table 1 moved to supplementary information as “Table A1”.
“ARMP-F14 was used to examine the effect of pressure on the oxygen diffusivity and the oxidation reactions in stabilization, and ARMP-F10 was used to prepare the carbonized and graphitized fibers” Why didn’t the authors use the same diameter for the both?
Answer:
Thank you for kind comment. We have been tried the same and similar experiments for long time over 30 years in our research group. From our experiment, it is very difficult to obtain the clear images of skin-core texture with the carbon fiber of diameter of less than 10 μm, even if we carefully controlled the stabilization conditions. Also, the fiber with diameter of over 10 μm usually showed the moderate mechanical properties. From these reasons, we classified the fiber diameter with 14 and 10 μm as a test fiber for texture and mechanical property, respectively.
“Figure 3 shows a schematic of MSB-TG-1300.” –Please consider moving the schematic to supplementary information.
Answer: Figure 3 moved to supplementary information as “Figure A1”.
“ARMP-F10 and ARMP-F14 had very similar profiles of oxygen uptake, which means that they experienced almost the same oxidation reactions under the same applied pressure.” – Did the authors expect any significant change in oxygen uptake since the diameter difference between these two samples was very close? Could you please explain why such close fiber diameter difference was selected for this study?
Answer:
Thank you for a valuable comment. As the review commented, the oxidation reactions should be similar on the two pitch fibers, however, such same or similar oxidation reactions could cause the very different oxidation state on the resulted stabilized fibers with different diameters. That is, the optimization condition (for example, optimized oxygen weight gain, homogeneous oxygen distribution, and so on) should be different on the fibers with different diameter. For the exact extraction of such an optimization condition for each condition, of course, we need too many experimental data. In this paper, we only tried to show the effectiveness of the oxidation stabilization under pressure for shortening the stabilization time. Problems of economic process should be issued, however, it must be review again for the revolution of the industry of the pitch based carbon fiber.
Table 3 and 4 provide a lot of data, however, it is hard to follow any trend. I am recommending obtaining some graphs out of these tables showing the significant trends only and to move the tables in the supplementary information for further reader’s interest.
Answer: We moved the tables in the supplementary information (Table A2 and A3) for further reader’s interest. Table 2 and 3 showed significant trends of mechanical properties and carbonization and graphitization yield of ARMP-CF10 and ARMP-GF10.
Page 12 (395-407): Here authors provide observations on the mechanical performance of the fibers without much insight or explanation. Please explain the trends as can be seen in the tables including the mechanism.
Answer:
Thank you for kind comment. I think the reviewer’s comment is right. As the reviewer’s comment, it is not enough the explanation of the revelation of the better mechanical properties of the resulted carbonized and graphitized fibers. Our answers were only concentrated on the homogeneous distribution and not excessed of oxygen weight gains of the oxidation stabilized fibers were easier to obtain under the air flow of the pressurized state than the stabilization under the air flow of the atmospheric one without deteriorating the mechanical properties of the resulted fiber. That is, our goal is the shortening the stabilization time in the air stabilization process and the proof of the effectiveness of the pressurized air flow on shortening the stabilization time. In the next time, we will challenge to elucidate the clear explanation of the reason for the improvement of the mechanical properties with more experimental results.
“From these results, stabilization under air pressures of 0.5 MPa cause homogeneous oxidation and increases mechanical properties more than other graphitized fibers. In contrast, the mechanical properties of ARMP-GF10 stabilized under air pressures of the atmospheric level and 1.0 MPa may decrease due to deficient and excess oxygen uptake in the center and outer parts of pitch fibers, respectively.” add to Line 398.
Reviewer 4 Report
Shimanoe et. al. studied that the oxidation-stabilization using pressurized air flows of 0.5 and 1.0 MPa could successfully shorten the total stabilization time to less than 60 min for manufacturing mesophase pitch-based carbon fibers without deteriorating mechanical performance.
Before publication of this manuscript, the following concerns should be addressed.
Abstract should be more quantitative and technical. Check it. Some characterization results are needed. Introduction should be updated by 2017, 2018 and 2019 published papers. Please check it and modify. Digital image of fabricated materials should insert. Line 284-288, needs references. Check it and modify. Please, cite a paper Carbohydrate polymers 211, 181-194 regarding fibers or in tensile part. Conclusion should be technical.Author Response
Thank you for your letter and the reviewers’ comments concerning our manuscript. Those comments are very much valuable and helpful for revising and improving our paper, as well as important for our further research. We have studied the comments carefully and have made the corrections. The responds to the reviewers’ comments are listed as follows:
Our response (in regular type) to the comments (in italics) follows:
Abstract should be more quantitative and technical. Check it. Some characterization results are needed.
Answer:
Thank you for kind comment. The amount of oxygen uptake might be a very specific according to the kind of mesophase pitch, not absolute one. From the reason, we did not use the amount of oxygen uptake. Some analytic estimation results such as SEM-EPMA have been additionally expressed to the abstract.
“The higher oxygen diffusivities resulted in a more homogeneous distribution of oxygen weight uptake across the transverse section of mesophase pitch fibers and allowed the improvement of the mechanical properties.”
Introduction should be updated by 2017, 2018 and 2019 published papers. Please check it and modify.
Please, cite a paper Carbohydrate polymers 211, 181-194 regarding fibers or in tensile part.
Answer:
We add to references “Yadav, M.; Chiu, F.C. Cellulose nanocrystals reinforced κ-carrageenan based UV resistant transparent bionanocomposite films for sustainable packaging applications. Carbohydrate Polymers 2019, 211, 181-194.”.
Digital image of fabricated materials should insert.
Answer:
SEM images of ARMP-SF14 and ARMP-GF14 insert to the results as Fig 6 and 7. And SEM images of ARMP-F14 are similar to ARMP-SF14. It is very difficult to obtain the clear images of skin-core texture with the carbon fiber of diameter of less than 10 μm, even if we carefully controlled the stabilization conditions.
Conclusion should be technical.
Answer:
We tried to revised according to your comment as possible as we could.
Round 2
Reviewer 4 Report
Accepted
Author Response
thank you very much for your comments and considering.